# Enhancement of a Graphene-Based Near-Field Thermophotovoltaic System by Optimization Algorithms and Dynamic Regulations

**Yitao Sheng**

CW Chu College, Jiangsu Normal University, Xuzhou 221116, China; 3020201494@jsnu.edu.cn

**Abstract:** Thermophotovoltaics (TPVs), a heat recovery technique, is faced with low efficiency and power density. It has been proven that graphene helps add new functionalities to optical components and improve their performance for heat transfer. In this work, I study Near-field radiative heat transfer in TPVs based on a composite nanostructure composed of Indium Tin Oxide (ITO) sheet and a narrow bandgap photovoltaic cell made from Indium Arsenide (InAs). I introduce a new way to calculate nonradiative recombination (NR) and compare the performance with and without the NR being considered. By comparing graphene modulated on the emitter (G-E), on the receiver (G-R), and on both the emitter and the receiver (G-ER), I find the G-ER case can achieve the highest current density. However, constrained by the bandgap energy of the cell, this case is far lower than the G-E case when it comes to efficiency. After applying variant particle swarm optimization (VPSO) and dynamic optimization, the model is optimized up to 43.63% efficiency and 11 W/cm$^2$ electric power at a 10 nm vacuum gap with a temperature difference of 600 K. Compared with before optimization, the improvement is 8.97% and 7.2 W/cm$^2$, respectively. By analyzing the emission spectrum and the transmission coefficient, I find that after optimization the system can achieve higher emissivity above the bandgap frequency, thus achieving more efficient conversion of light to electricity. In addition, I analyze the influence of temperature difference by varying it from 300 K to 900 K, indicating the optimized model at a 900 K temperature difference can achieve 49.04% efficiency and 52 W/cm$^2$ electric power. By comparing the results with related works, this work can achieve higher conversion efficiency and electric power after the optimization of relevant parameters. My work provides a method to manipulate the near-field TPV system with the use of a graphene-based emitter and promises to provide references in TPV systems that use low bandgap energy cells.

**Keywords:** near-field radiative heat transfer; graphene; dynamic optimization; particle swarm optimization; thermophotovoltaic system

## 1. Introduction

Global energy consumption is increasing year by year, especially when all kinds of waste heat and waste gas are produced by industrial energy consumption. The waste heat of other processes can be used as a thermal emission source, in which case thermophotovoltaic (TPV) systems have been considered as one of the most promising methods for recycling waste heat [1–5]. The main challenge of the TPV system is to elevate the efficiency and the current density. However, the efficiency of a TPV system is inherently limited by the spectral emissivity of the emitter—the blackbody limit of Planck's law, since only propagating waves contribute to energy transfer [6–19]. In the near field, where the gap between two objects is less than their Wien wavelength, the radiative heat transfer between them can exceed the blackbody limit. The near-field radiative heat transfer contributed from evanescent waves can exceed that predicted by the Stefan–Boltzmann law by 1~3 orders of magnitude, especially when the surface plasmon polaritons (SPP) or surface phonon polaritons (SPhP) are excited [20–23]. Due to the SPP and SPhP, a near-field

thermophotovoltaic (NFTPV) system can achieve a higher power density and a higher efficiency than a far-field system [24–27], making it possible to recover waste heat.

The temperature difference and match between the frequency of surface polaritons supported by the emitter and the bandgap of the cell should be taken into consideration in order to improve the efficiency of the NFTPV system conversion. To make the system more applicative, the operating temperature of a TPV receiver is always set to near room temperature and the emitter temperature is adjustable. Graphene is widely used in nano materials owing to its striking optical and electrical properties [28]. Recently, it has been experimentally proven that graphene plays an important role in improving the performance of NFTPV systems [29,30]. Moreover, Beiranvand et al. have shown that modulating graphene on the optical modules and controlling the chemical potential of the graphene layer promises to achieve spectral selective tunable absorption and thus can be used in a wide range of applications from photo-detectors to multi-spectral tunable reflectors [31,32].

In the context of heat transfer, the special optical properties of plasmons in graphene promises to improve the performance of the thermodynamic component when covered with a layer of graphene sheet by elevating its dielectric for radiative heat [33–35]. The thermally excited plasmons contributed by the graphene can modulate the heat flux by electrical means, as it has excellent tunability from near-infrared to terahertz frequencies, which achieves a rapid regulation of the heat flux [36,37].

In this paper, I consider four cases to compare the effectiveness of graphene in NFTPV systems: no graphene cover (G-N case), graphene on the emitter (G-E case), graphene on the receiver (G-R case), and graphene on both the emitter and the receiver (G-ER case). I use Indium Tin Oxide (ITO) as the emitter due to its high-temperature stability and its wide application. ITO, a degenerate semiconductor and electrically conductive material which is practically transparent in the visible spectrum [38], performs well within the system. Meanwhile, the plasma frequency of ITO can be easily adjusted by controlling the oxygen content in the deposition process, making it possible to dynamically optimize the NFTPV system.

Indium Arsenide (InAs) cells are widely used in modulating NFTPV systems, but modulation by combining graphene and an InAs system is scarcely studied. Considering the fact that the bandgap energy of InAs is 0.354 eV higher than InSb whose bandgap energy is 0.17 eV at 300 K, and the external quantum efficiency of the InAs is lower than that of an InSb cell, studies on focusing on improving the efficiency of the InAs cell are meaningful for further developing the practical application of the InAs cell. Thus, in this paper, I use the narrow-bandgap thin film InAs cell as the receiver [39,40]. The InSb-based NFTPV system is also expected to be effective, owing to its low bandgap energy.

Huang et al. calculated the efficiency of the NFTPV system using an InAs cell and ITO emitter, which could reach nearly 53% and yielded an electric power density of 100 W/cm$^2$ at 1600 K [41]. However, this result is idealized, because they did not take the nonradiative recombination (NR) of the cell into consideration, and the operating temperature was very high, which may have led to the waste of the heat source and is difficult to achieve while in practical use. Since the NR is crucial in assessing the performance of the system in actual situations, it is important to take the NR in consideration while assessing. Zhao et al. considered the NR of the cell [42], but in that model, the function of the applied voltage on the cell was not explained clearly; this is possibly not applicable when the applied voltage is relatively high, e.g., when the applied voltage approaches the open circuit voltage. After considering the four cases above, I analyzed the system performance through comparing the performance with and without the NR included, at different applied voltages between the emitter and the receiver, different emitter thicknesses, and different temperatures.

In consideration of the background above, the performance of an NFTPV system with graphene is investigated in this paper. After considering the position of the graphene in the system, I analyze the performance of the system at different applied voltages, thicknesses and plasma frequencies of the emitter, and differing chemical potential of the graphene and working temperatures of the emitter. My work is organized as follows: in the following

section, an NFTPV system is proposed. By taking it as an example, I build a model to actively optimize the performance of the system. In Section 3, by evaluating the difference in performance when NR is considered or not, the need to consider the NR is demonstrated. After that, four cases of the position of the graphene are analyzed to further illustrate the effect of the graphene. Moreover, variant particle swarm optimization (VPSO) is employed to optimize the thickness and plasma frequency of the emitter. In addition, I vary the chemical potential of the graphene to further promote the effect of the graphene on the NFTPV system by dynamic optimization analysis. Finally, by altering the working temperature difference to highlight the effect of temperature on the system and comparing the results with related works, the contribution of my work promises to provide further insight into graphene-based NFTPV systems.

## 2. Materials and Methods

In this section, I will introduce the theory and methods regarding the graphene-based NFTPV system. Figure 1 shows the schematic diagram of four graphene-based cases of my model. The thickness and temperature of the ITO emitter and the InAs cell are, respectively, $t_1$ and $T_1$, $t_2$ and $T_2$. The vacuum gap between the emitter and the cell is denoted by d. In this work, the temperature of the InAs is assumed to be 300 K. The chemical properties of ITO are unstable when the temperature exceeds 1700 K, so its upper limit is set to 1200 K to ensure its stability, and its lower limit is set to 600 K to strike a larger temperature difference between the emitter and the receiver, so as to achieve higher system efficiency. However, in the near-field heat transfer process, the photons with less energy than the bandgap of the cell cannot be converted into electricity, but only into heat, which causes the operating temperature of the cell to rise. Note that the conversion efficiency of the cell will drastically decrease when the operating temperature is very high, e.g., higher than 330 K. Additionally, a high operating temperature may also burn out the cell. Thus, the heat sink here helps cool the cell to ensure the cell works stably.

I calculate and assess the performance of the NFTPV system depicted in Figure 1 in two cases: NR-excluded and NR-included. For the two cases above, their efficiency and electric power at different voltages should be compared. NR inside the cell can be obtained by [43]

$$J_{nrad} = q \times t_2 \times (C_n + C_p) \times n_i{}^3 \times \exp(\frac{3qV}{2k_B T_2}) \tag{1}$$

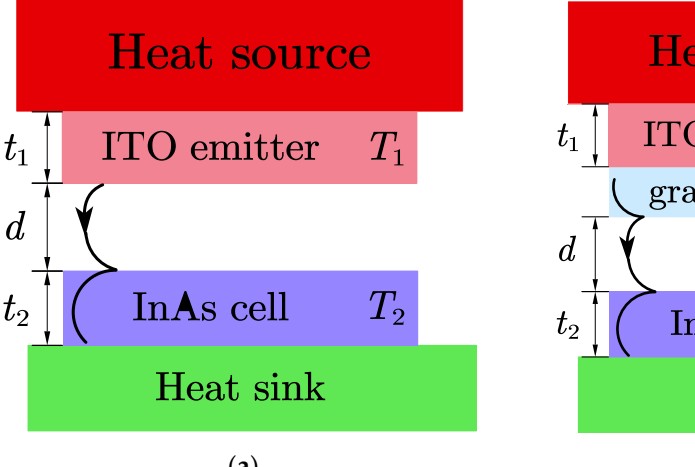

(a)          (b)

**Figure 1.** *Cont.*

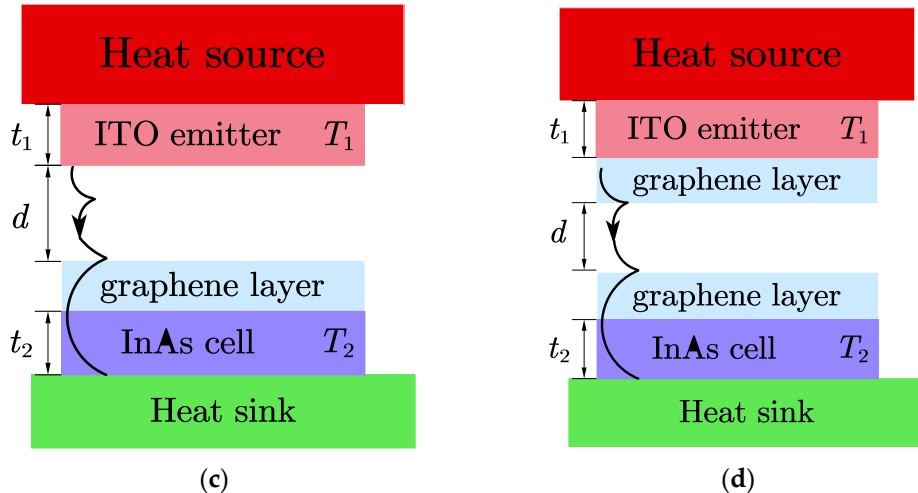

**Figure 1.** Schematic diagram of the NFTPV system. (**a**) G-N case. (**b**) G-E case. (**c**) G-R case. (**d**) G-ER case. The thickness and temperature of the ITO emitter and the InAs cell are, respectively, $t_1$ and $T_1$, $t_2$ and $T_2$. The vacuum gap between the emitter and the cell is denoted by $d$.

In Equation (1), $n_i$, $k_B$, $q$ stand for the intrinsic carrier concentration, the Boltzmann coefficient, elementary charge, respectively. $C_p$ and $C_n$ are the Auger recombination coefficients. Here, I take the parameters from Ref. [42]: $n_i = 6.06 \times 10^{14} \mathrm{cm}^{-3}$, $C_n = C_p = 2.26 \times 10^{-27} \mathrm{\ cm}^{-6} \cdot \mathrm{s}^{-1}$. The major difference between the NR-included and the NR-excluded cases lies in the power density of the system. The current density of the two cases in the cell can be obtained by

$$J_{NR-in} = J_{ph} - J_{rad} - J_{nrad} \tag{2}$$

$$J_{NR-ex} = J_{ph} - J_{rad} \tag{3}$$

In Equations (2) and (3), $J_{ph}$ stands for the photogenerated current density and $J_{rad}$ stands for the radiative recombination of the cell. $J_{ph}$ and $J_{rad}$ can be obtained by

$$J_{ph} = q \times \int_{\omega_c}^{\infty} \frac{\Theta(\omega, T_1, 0)}{4\pi^2 \hbar \omega} [\int_0^{\infty} \xi(\omega, \beta) \beta d\beta] d\omega \tag{4}$$

$$J_{rad} = q \times \int_{\omega_c}^{\infty} \frac{\Theta(\omega, T_2, V)}{4\pi^2 \hbar \omega} [\int_0^{\infty} \xi(\omega, \beta) \beta d\beta] d\omega \tag{5}$$

In Equations (4) and (5), $\hbar$ is the reduced Planck constant, $V$ is the applied voltage of the cell, $\omega$ is the angular frequency, and $\beta$ is the transverse wavevector. $\xi(\omega, \beta)$ is the energy transmission coefficient (also called photon tunneling probability) [44,45]. $\omega_c$ is the angular frequency corresponding to the bandgap of the cell, and the bandgap energy of the InAs is 0.354 eV at 300 K. Note that the emission spectrum can be calculated by integrating the energy transmission coefficient only over $\beta$. In addition, $\Theta(\omega, T_k, V_k)$ is the mean energy of the Planck harmonic oscillator at angular frequency $\omega$ [46], which can be calculated by

$$\Theta(\omega, T, V) = \frac{\hbar \omega}{\exp(\frac{\hbar \omega - qV}{k_B T}) - 1} \tag{6}$$

The efficiency ($\eta$), output power density ($P$), and current density ($J$) are always regarded to be an important embodiment of the system performance, and they are targets to consider when optimizing. The output power density $P$ of the cell is calculated by $P = JV$. Thus, when the calculation includes the NR of the cell, $P$ can be calculated as $P = J_{NR-in}V$,

and when the calculation excludes that, $P$ can be calculated as $P = J_{NR-ex}V$. The efficiency of the system is obtained by

$$\eta = P/E \times 100\% \tag{7}$$

where $E$ represents the radiative heat flux between the cell and the emitter. Additionally, the radiative heat flux can be divided into two parts: frequency above the bandgap due to electronic excitations denoted by $E^e$ and frequency below the bandgap due to phonon-polariton excitations denoted by $E^p$, i.e., $E = E^e + E^p$, and

$$E^e = \frac{1}{4\pi^2} \int_{\omega_c}^{\infty} [\Theta(\omega, T_1, 0) - \Theta(\omega, T_2, V)][\int_0^{\infty} \xi(\omega, \beta)\beta d\beta]d\omega \tag{8}$$

$$E^p = \frac{1}{4\pi^2} \int_0^{\omega_c} [\Theta(\omega, T_1, 0) - \Theta(\omega, T_2, 0)][\int_0^{\infty} \xi(\omega, \beta)\beta d\beta]d\omega \tag{9}$$

The energy transmission coefficient is contributed by both the transverse electric (TE) waves (s-polarization) and transverse magnetic (TM) waves (p-polarization), i.e., $\xi(\omega, \beta) = \xi_s(\omega, \beta) + \xi_p(\omega, \beta)$, and

$$\xi_j(\omega, \beta) = \begin{cases} \frac{\left(1-|R_{1j}|^2\right)\left(1-|R_{2j}|^2\right)}{\left|1-R_{1j}R_{2j}e^{2ik_{z1}d}\right|}, & \beta < k_0 \\ \frac{4\left[Im(R_{1j})Im(R_{2j})\right]e^{-2|k_{z1}|d}}{\left|1-R_{1j}R_{2j}e^{2ik_{z1}d}\right|^2}, & \beta > k_0 \end{cases} \tag{10}$$

with $j$ denoting s or p polarization. In Equation (10), $k_0$ and $k_{z1}$ represent the magnitude and the z-component wavevector, here $k_0 = \omega/c_0$ where $c_0$ stands for the speed of light and $k_{z1} = \sqrt{k_0^2 - \beta^2}$. Note that only the propagating waves ($\beta < k_0$) contribute to the heat transfer in the far-field system, whereas in the near-field system, the evanescent waves ($\beta > k_0$) dominate the heat transfer. $R_{1j}$ and $R_{2j}$, respectively, represent the reflection coefficient of the emitter and the receiver; when the emitter is covered with a single layer of graphene, $R_{1j}$ and $R_{2j}$ take the following forms [47–50]:

$$R_{l,p} = \frac{r^p{}_{12,l} + \left(1 - r^p{}_{12,l} - r^p{}_{21,l}\right) \times r^p{}_{23,l} \times \exp(2 \times i \times k^p{}_{z2,l} \times t_l)}{1 - r^p{}_{12,l} \times r^p{}_{23,l} \times \exp(2 \times i \times k^p{}_{z2,l} \times t_l)} \tag{11}$$

$$R_{l,s} = \frac{r^s{}_{12,l} + \left(1 + r^s{}_{12,l} + r^s{}_{21,l}\right) \times r^s{}_{23,l} \times \exp(2 \times i \times k^s{}_{z2,l} \times t_l)}{1 - r^s{}_{21,l} \times r^s{}_{23,l} \times \exp(2 \times i \times k^s{}_{z2,l} \times t_l)} \tag{12}$$

where $l$ denotes 1 or 2.

The wavevector can be split into two cases: p-polarized and s-polarized. The p-polarized wavevector of the body $l$ is $k^p{}_{z2,l} = \sqrt{\varepsilon_{l,\perp}k_0^2 - \varepsilon_{l,\perp}\beta^2/\varepsilon_{l,\parallel}}$, and the s-polarized wavevector of that is $k^s{}_{z2,l} = \sqrt{\varepsilon_{l,\perp}k_0^2 - \beta^2}$. When in the vacuum gap, I assume $k_{z3} = k_{z1}$. Moreover, $r^p{}_{ab,l}$ and $r^s{}_{ab,l}$ can be calculated by [50–52]

$$r^p{}_{ab,l} = \frac{k^p{}_{za,l} \times \varepsilon_{a,\perp} - k^p{}_{zb,l} \times \varepsilon_{b,\perp} + \frac{k^p{}_{za,l} \times k^p{}_{zb,l} \times \sigma}{\varepsilon_0 \times \omega}}{k^p{}_{za,l} \times \varepsilon_{a,\perp} + k^p{}_{zb,l} \times \varepsilon_{b,\perp} + \frac{k^p{}_{za,l} \times k^p{}_{zb,l} \times \sigma}{\varepsilon_0 \times \omega}} \tag{13}$$

$$r^s{}_{ab,l} = \frac{k^s{}_{za,l} - k^s{}_{zb,l} - \sigma \times \omega \times \mu_0^2}{k^s{}_{za,l} + k^s{}_{zb,l} + \sigma \times \omega \times \mu_0^2} \tag{14}$$

In Equations (13) and (14), $a$ and $b$ can be 1, 2, or 3. For regions 1 and 3, $\varepsilon_{\perp}$ and $\varepsilon_{\parallel}$ are the vertical and parallel components of the dielectric function, and for region 2, the vacuum layer, I assume $\varepsilon_2 = \varepsilon_{\perp} = \varepsilon_{\parallel} = 1$. The permittivity of ITO shows isotropy in the x-y plane. Thus, I can conclude $\varepsilon_1 = \varepsilon_{1,\perp} = \varepsilon_{1,\parallel}$. The dielectric function of ITO can be explained by a free-electron Drude model: $\varepsilon(\omega) = \varepsilon_{\infty}\left(1 - \omega_p^2/(\omega^2 + i\omega\Gamma)\right)$, where $\varepsilon_{\infty} = 4$,

$\omega_p$ = 0.4–0.9 eV, and $\Gamma$ = 0.1–0.15 eV [53]. Furthermore, $\sigma$ stands for the conductivity of the graphene, which can be calculated by

$$\sigma = \frac{q^2 \times \mu \times \tau}{\pi \times \hbar^2 (1 - i\omega\tau)} \tag{15}$$

I assume $\Gamma$ = 0.1 eV, $\omega_p$ = 0.4 eV at first, and the optimal plasma frequency is selected through stepwise optimization. The surface plasmonic resonance is strong on the interface of the emitter and the vacuum layer near the surface resonance frequency $\omega_{res}$ [41]. Furthermore, I need to ensure the $\omega_{res}$ is above the bandgap to improve the performance of the system.

## 3. Results and Discussion

### 3.1. The Performance of Four Graphene-Based Cases

In order to analyze the respective performance of the NR-included and the NR-excluded cases at different voltages, I firstly set $t_1$ = 30 nm, $t_2$ = 390 nm, $T_1$ = 900 K, $T_2$ = 300 K, and $d$ = 10 nm in this paper. As can be seen in Figure 2a, if the NR is excluded, the efficiency of the NFTPV system will be nearly 15% higher than that of the actual situation, with the actual situation reaching 34.66 % at 0.2 V and the ideal situation reaching 39.87 % at 0.225 V. In Figure 2a,b, the two function curves of the NR-included and the NR-excluded cases are essentially overlapping when the applied voltage is less than 0.15 V, indicating that when the applied voltage is relatively low, whether the NR is taken into consideration or not, it does not exert a huge influence to the assessment. However, the applied voltage on the InAs cell always needs to be higher than 0.2 V when the conversion efficiency is high. Thus, the NR of the cell is nonnegligible in my discussion for further application.

In the model mentioned above, the graphene layer can be modulated on the emitter side, the receiver side, and both the emitter and the receiver side in addition to no graphene being modulated. From Figure 2d,e, it can be seen that it is the G-ER case that achieves the highest current density and electric power compared with the other cases. In the G-ER case, one layer of graphene above the emitter and the receiver can absorb the majority of incoming radiation from the heat source and the emitter, respectively, ensuring and helping the excitation of plasmons in graphene to automatically tune in resonance with the emitted light in the midinfrared range and the frequency of the surface mode in the opposing body, thus resulting in a significant increase in the current density and the electric power. The result is essentially consistent with Ref. [54] where the G-N case can enhance the electric power and the current density of the system. Nevertheless, the efficiency defined in the article only considers the energy converted into electricity. Actually, the TPV efficiency should take into account not only the energy converted into electricity, but also the energy converted into heat.

In Figure 2c, the efficiency of G-ER case and G-R case is lower than 5 %. This is in sharp contrast to the G-N case, which can be explained by Figure 2f. Although the two cases can reach high current density and electric power, the energy of photons emitted by the emitter is relatively low, i.e., lower than the bandgap energy of the InAs cell. Hence, the majority of the photons are not able to be converted into electricity and represent a massive waste of heat, and that is why the efficiency defined in Ref. [54] is higher than that in the G-N case. Moreover, from Figure 2c–f, it can be seen that the G-E case can not only reach high current density and electric power of approximately 25% lower than that of the G-ER case, but also ensures high efficiency, reaching 40.84 % at 0.21 V. In my discussion below, I choose the G-E case to further clarify the theory. Note that owing to the majority of photons emitted by the emitter remaining at a low level of energy, if a lower bandgap energy cell is applied, i.e., the bandgap energy is much lower than the mean value of these photons, the G-ER case is worth considering and we can further consider the multilayer graphene structure.

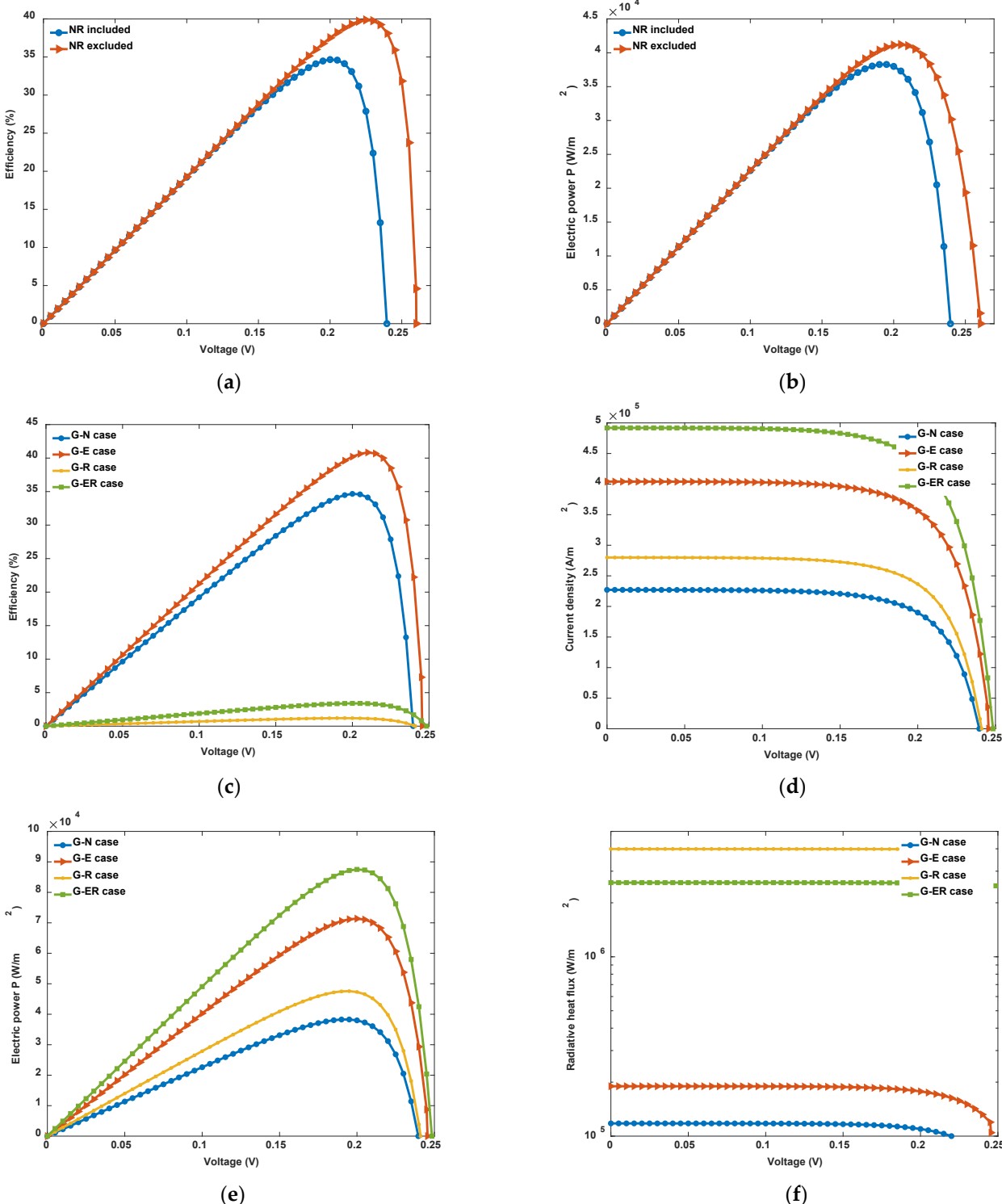

**Figure 2.** The (**a**) efficiency, (**b**) electric power at different applied voltages comparing NR-included and NR-excluded at $t_1 = 30$ nm, $t_2 = 390$ nm, $T_1 = 900$ K, $T_2 = 300$ K and $d = 10$ nm. The (**c**) efficiency, (**d**) current density, (**e**) electric power, (**f**) radiative heat flux at different applied voltages of four cases of placements of graphene at $t_1 = 30$ nm, $t_2 = 390$ nm, $\mu = 0.3$ eV, $T_1 = 900$ K, $T_2 = 300$ K, and $d = 10$ nm.

### 3.2. ITO Thickness and Plasma Frequency Optimized by Particle Swarm Optimization

In terms of the independence between the ITO thickness and plasma frequency, a multi-objective variant particle swarm optimization (VPSO) model can be built to improve

the performance of the NFTPV system [55,56]. Considering the dimensions among them are different, it is important to eliminate the dimensions by $\phi_i = \frac{\zeta_i}{\zeta_{i,max}}$, where $I$ = 1, 2, 3, respectively, denote efficiency, current density, and electric power. In addition, $\zeta_{i,max}$ denotes the result calculated in the G-E case. By listing the practical limitations of the thickness and the plasma frequency, the target can be described as follows:

$$max \sum_{i=1}^{3} \lambda_i \phi_i \tag{16}$$

$$s.t. \begin{cases} 10\,\text{nm} \leq t_1 \leq 100\,\text{nm} \\ 0.4\,\text{eV} \leq \omega_p \leq 0.9\,\text{eV} \\ \sum_{i=1}^{3} \lambda_i = 1 \end{cases} \tag{17}$$

where, generally, $\lambda_i$ ($i$ = 1, 2, 3) can be set as $\frac{1}{3}$ in order to simplify the solution procedure. The schematic diagram of the VPSO is depicted in Figure 3. By setting the number of particles, number of variables, and the variable coefficient to 30, 2, and 0.03, respectively, the optimization of the VPSO algorithm is realized by MATLAB (MathWorks, Natick, the USA). Codes can be seen in the Supplementary Materials.

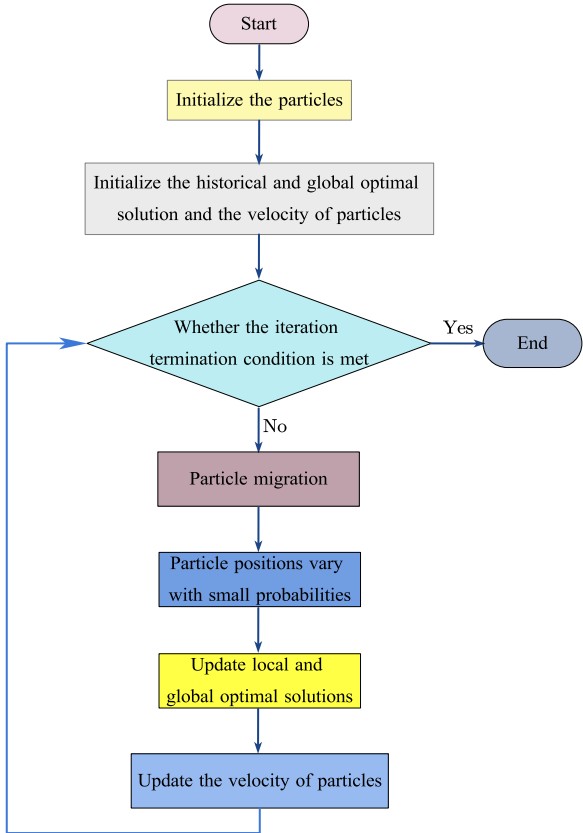

**Figure 3.** The schematic diagram of the VPSO by setting the number of particles, the number of variables, and the variable coefficient to 30, 2, and 0.03, respectively.

According to the model, the global optimal solution is $t_1$ = 100 nm, $\omega_p$ = 0.5 eV. One knows that near the surface plasmonic frequency, there is a strong surface plasmonic resonance on the interface of the graphene-covered ITO and the vacuum. When the plasma frequency of the ITO film reaches 0.5 eV, the surface plasmonic resonance is strong near the frequency $\omega_{res}$, which is higher than the bandgap energy of the InAs cell. Thus, most of the photons can be converted into electricity by the cell. As is shown in Figure 4, by VPSO, the efficiency peaks at 42.93% at 0.215 V and the short-circuit current density reaches 58 A/cm$^2$.

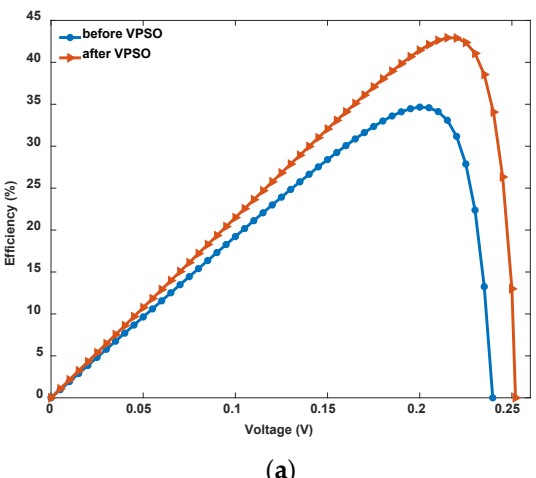
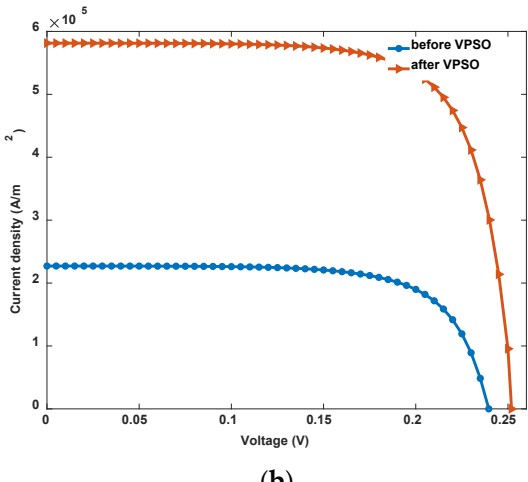

**(a)**　　　　　　　　　　　　　　　　　　　　　　**(b)**

**Figure 4.** The (**a**) efficiency and (**b**) current Density at different applied voltages before and after VPSO.

### 3.3. Performance of Different Chemical Potentials of the Graphene

In this part, I analyze the influence of chemical potential of the graphene layer on the NFTPV system. For the graphene, its conductivity is assumed to be independent of the in-plane wavevector and composed of the intraband and interband conductivity. I consider the chemical potential to be 0.1 eV, 0.3 eV, 0.5 eV, 0.7 eV, respectively, and compare the performance of the system for each, which can be seen in Figure 5. In Figure 5a–d, the $\mu = 0.1$ eV situation is much better in that not only is the output power density of the cell increased, but also the radiative heat flux between the cell and the emitter is promoted. Thus, in the discussion below, the chemical potential $\mu$ is decreased to 0.1 eV to optimize the performance of the whole system. In fact, decreasing the chemical potential to a proper level may result in a larger heat transfer rate when the emitter temperature is very high, e.g., higher than 900 K. Moreover, since the high-frequency polaritons would be more significant when the emitter temperature is higher than 900 K, it also indicates that we should alter the emitter temperature to further explore the performance of the NFTPV system.

### 3.4. Comparison between Performances before and after Optimization

After optimizing the performance of the NFTPV system, the results can be concluded as follows: the optimized chemical potential is 0.1 eV, the thickness of the ITO emitter is 100 nm, and the plasma frequency is 0.5 eV. Hence, I can compare the performance before and after my optimization, which is shown in Figure 6. From the bar chart, it can be seen that the current density after optimization increases by about 160% from 22.7 A/cm$^2$ before optimization to 59.7 A/cm$^2$, along with the efficiency and the open-circuit voltage, respectively, increasing by about 8.97% from 34.66% efficiency to 43.63% efficiency, and by 0.0118 V from 0.2396 V to 0.2514 V. Considering the bandgap of the InAs cell is 0.354 eV, the InAs frequency bandgap is about $5.37 \times 10^{14}$ rad/s. By integrating the transmission coefficient over $\beta$, the emission spectrum is given in Figure 6d. The spectral heat flux is much higher in the above-bandgap frequency range after optimization. The optimization promotes the performance of surface plasmon-photon polaritons in near-field heat transfer, indicating optimization of the graphene modulated on the emitter, and that the thickness and the plasma frequency of the ITO emitter are effective.

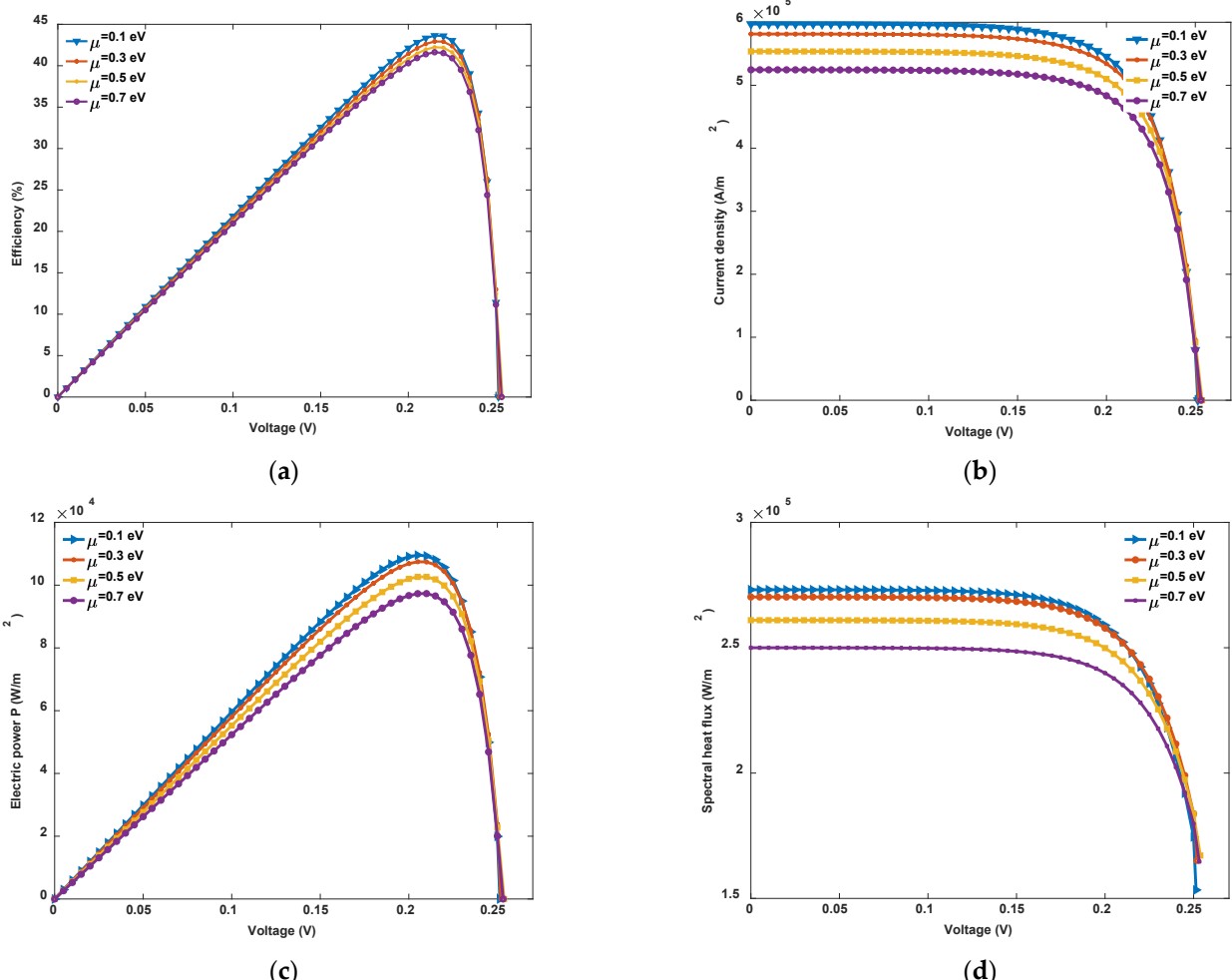

**Figure 5.** The (**a**) efficiency, (**b**) current density, (**c**) electric power and (**d**) radiative heat flux at different applied voltages of four potential chemicals at $t_1 = 100$ nm, $\omega_p = 0.5$ eV when the emitter is covered with a single layer of graphene sheet.

Furthermore, I calculate the energy transmission coefficient as a function of $\omega$ and $\beta$ for the NFTPV system before and after optimization when the vacuum gap d between the emitter and the receiver is 10 nm, which is shown in Figure 7. In the above-bandgap frequency range, InAs exhibits significant absorption as it is a direct bandgap material. Both the waveguide modes in the InAs film and the surface plasmon polariton on the interface of ITO and vacuum are excited as indicated by the bright bands. Only when the frequency of the photon is above the frequency bandgap can the photon be absorbed by the InAs cell and be translated into electricity, whereas if the frequency is below the bandgap, it can only be translated into heat, which contributes to the energy lost. From Figure 7, it can be seen that when the angular frequency is high (higher than bandgap frequency), the energy transmission coefficient after optimization is much larger than that before optimization, which means the performance of heat transfer and the conversion of light to electricity are obviously improved.

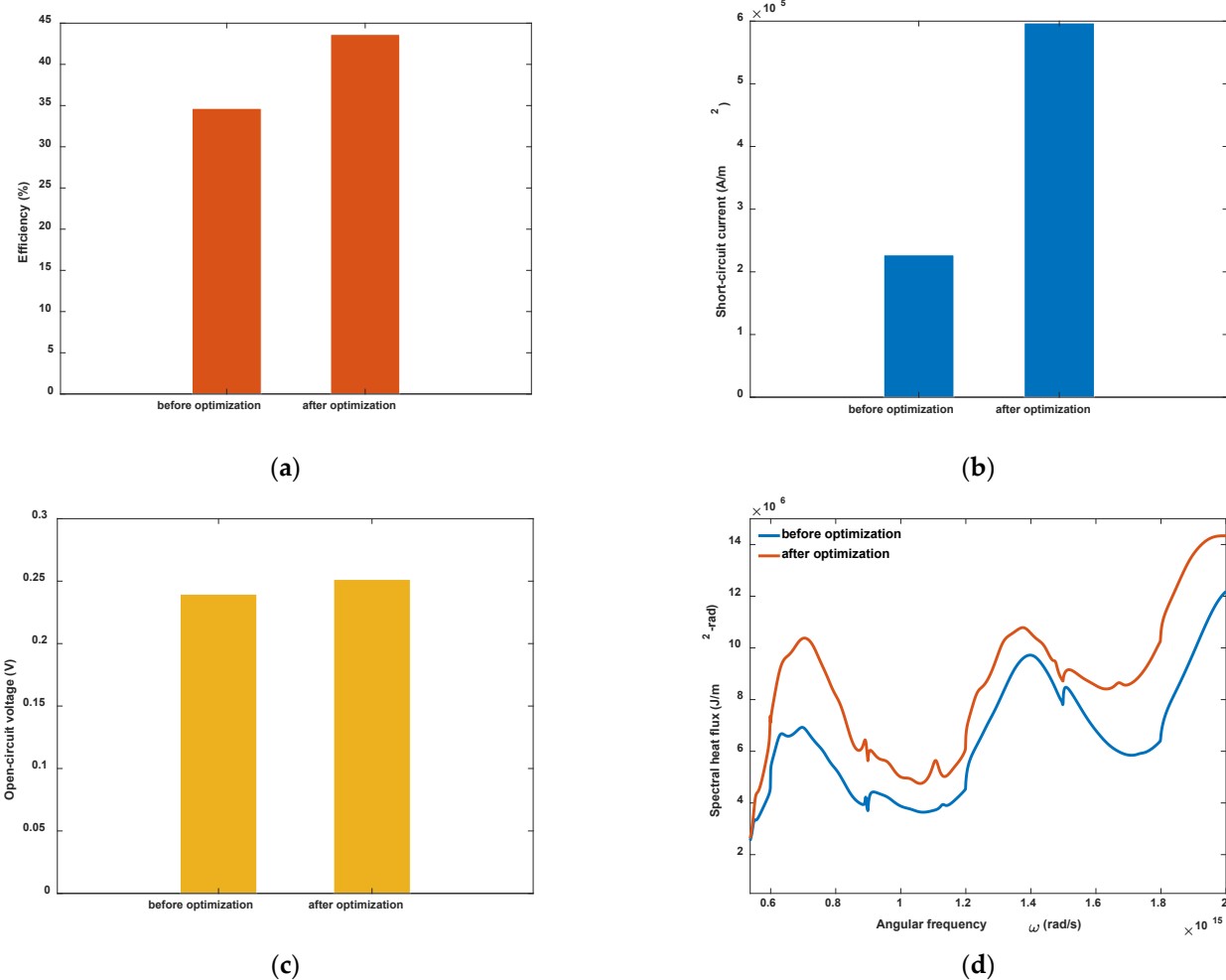

**Figure 6.** Comparison of the performance: (**a**) efficiency (**b**) short-circuit current density (**c**) open-circuit voltage (**d**) emission Spectrum before and after optimization before and after optimization.

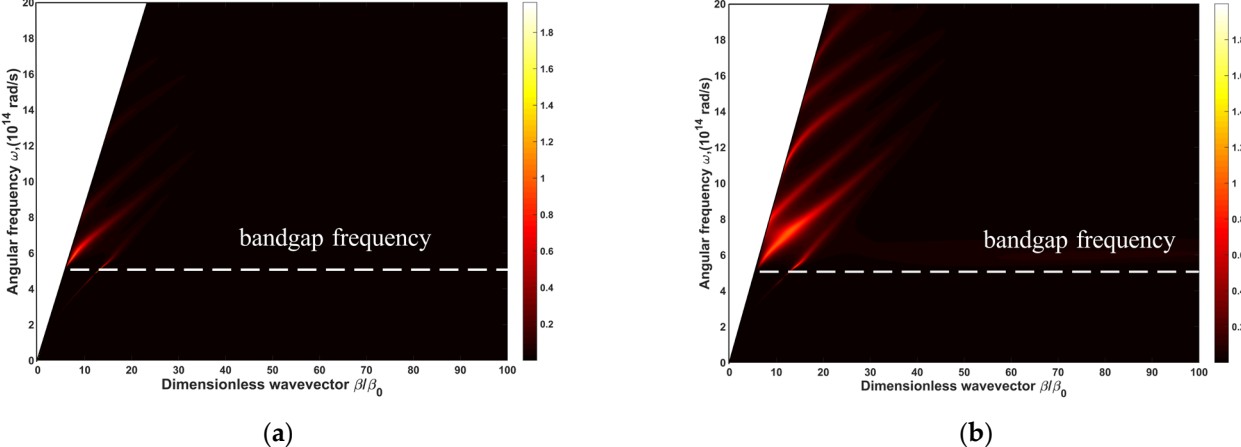

**Figure 7.** Comparison of the performance before (**a**) and after (**b**) optimization in energy transmission coefficient in a high-frequency spectral range at $t_1$ = 100 nm, $\mu$ = 0.1 eV, $\omega_p$ = 0.5 eV when the emitter is covered with a single layer of graphene sheet. The white dashed line is the bandgap frequency of the cell. Wavevector $\beta$ is normalized by $\beta_0 = \omega_0/c_0$ and $c_0$ the speed of light in vacuum.

### 3.5. Performance of Different ITO Emitter Temperatures

Based on the NFTPV model considered above, I propose a modification to the ITO emitter temperature to further improve the performance of the system. In the far-field TPV system, with increasing temperature difference, there is a sharp increase in the photocurrent density because more in-band photons are excited, which can be described by Planck's equation and the experimental effective emissivity of the thermal emitter. Now, I consider the near-field TPV system by varying the temperature difference from 300 K to 900 K. The result is shown in Figure 8. From Figure 8a, it can be seen that the efficiency peaks at 49.04% at 0.24 V at a 900 K temperature difference; by contrast, the maximum efficiency is only about 26.5% at 0.155 V at a 300 K temperature difference, and the open-circuit voltage increases from 0.195 V to 0.275 V. From Figure 8b–d, it can be seen that the increase in temperature difference leads to a sharp and striking improvement in current density, electric power and photon flux, i.e., two orders of magnitude, showing the system allows electricity generation with up to 49.04% efficiency, 52 W/cm$^2$ electric power, and 245 A/cm$^2$ current density at a 900 K graphene-based temperature difference at $d = 10$ nm. It shows high temperature difference is surely useful to improve the graphene-based NFTPV system. Extra discussion is needed to further optimize the performance when the temperature difference alters.

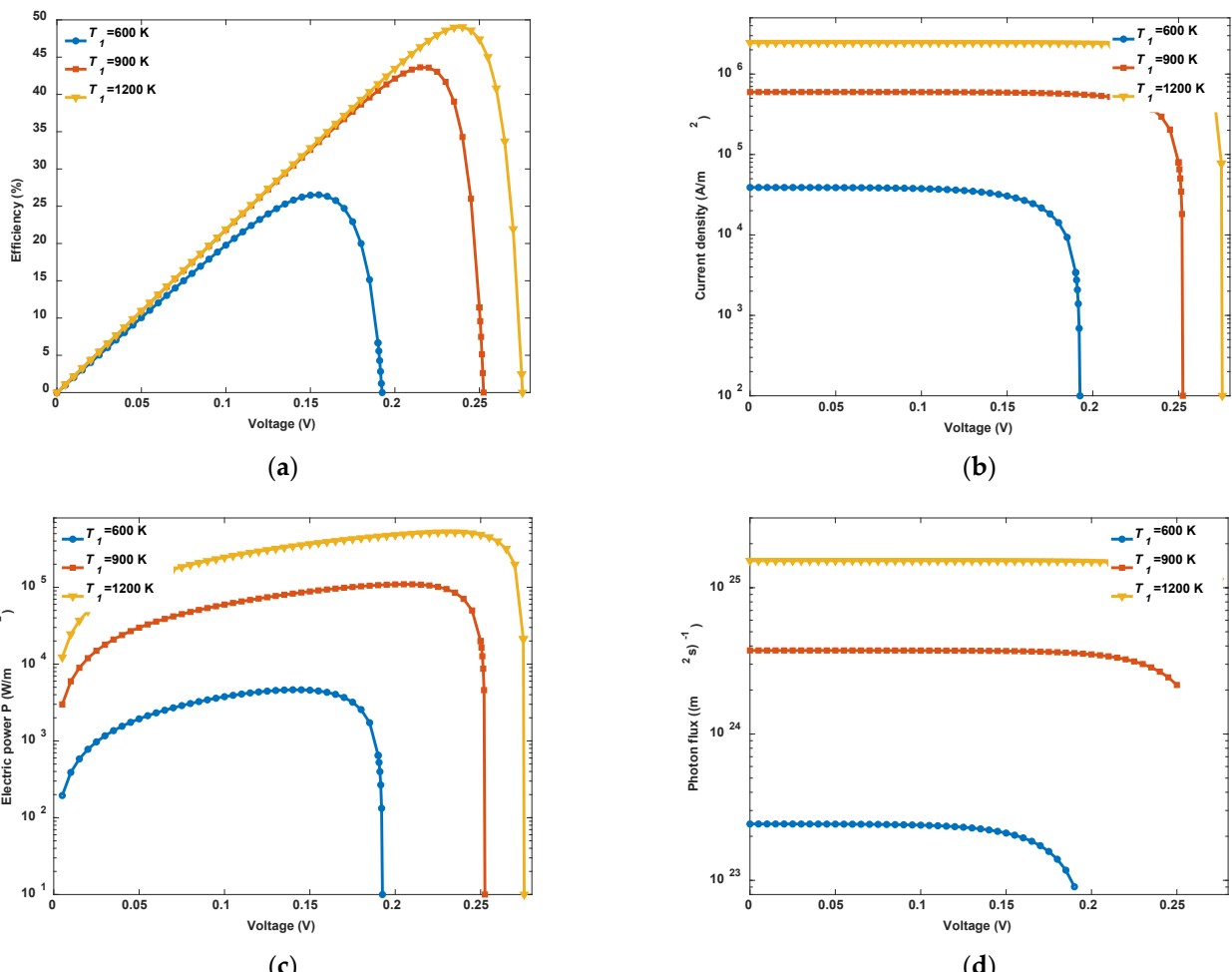

**Figure 8.** The (**a**) efficiency, (**b**) current density, (**c**) electric power, (**d**) photon flux transferred from the emitter and the receiver at different applied voltages and different ITO emitter temperatures at $t_1 = 100$ nm, $\mu = 0.1$ eV, $\omega_p = 0.5$ eV when the emitter is covered with a single layer of graphene sheet.

### 3.6. Comparison of the Results above with Related Works

Considering the fact that NR is often neglected in most articles, I list the result of NR-excluded and NR-included results and add some comparisons with related works at gap distance $d = 10$ nm in Table 1. Ref. [42] used the same devices as were used in this work and the authors achieved 39.30% efficiency and 11.0 W/cm$^2$ electric power. The conversion efficiency is lower than the result of this work. However, the electric power is close to the G-E case in this work. If one considers the G-ER case, the electric power can achieve 13.0 W/cm$^2$ after optimization, which is about 20 % higher than that of Ref. [41]. Ref. [57] demonstrated the Si-InGaAs system by introducing a pair of reflectors, reaching 52.00% efficiency when the temperature difference was 1100 K with NR excluded, which is lower than the result from this work when the temperature difference was 900 K with NR excluded. Ref. [27] studied an NFTPV system consisting of a tungsten-nanowire-based hyperbolic metamaterial emitter and an In$_{0.2}$Ga$_{0.8}$Sb cell. By optimizing the filling ratio of Al$_2$O$_3$, the maximum electric power peaked at 58.0 W/cm$^2$ thanks to the high temperature difference between the emitter and the receiver. Ref. [58] showed an NFTPV system with an ITO-coated tungsten emitter and an InAs cell. When taking NR into consideration, the authors achieved 39.80% efficiency.

**Table 1.** Comparison of the results in this work with related works at gap distance $d = 10$ nm.

| Structure | NR Included or Not | Efficiency (%) | Electric Power (W/cm$^2$) | Temperature Difference (K) | Ref |
|---|---|---|---|---|---|
| ITO InAs G-E | Yes | 43.63 | 11.2 | 600 | This work |
| ITO InAs G-E | No | 46.23 | 11.8 | 600 | This work |
| ITO InAs G-E | Yes | 49.04 | 52.0 | 900 | This work |
| ITO InAs G-E | No | 57.47 | 55.0 | 900 | This work |
| ITO InAs | Yes | 39.30 | 11.0 | 600 | [42] |
| Si InGaAs | No | 52.00 | 15.0 | 1100 | [57] |
| Tungsten In$_{0.2}$Ga$_{0.8}$Sb | No | 23.10 | 58.0 | 1700 | [27] |
| W-ITO InAs | Yes | 39.80 | 9.0 | 600 | [58] |

### 4. Conclusions

In this work, I investigate the performance of a graphene-based near-field TPV system by comparing graphene modulated on the emitter, on the receiver, and on both the emitter and the receiver after introducing a new method for calculating nonradiative recombination (NR) and comparing the performance when the NR is or is not considered. The thickness and the plasma frequency of the ITO emitter is optimized by variant particle swarm optimization; the chemical potential of the graphene is improved dynamically. I show that the NFTPV system can achieve 43.63% efficiency and 11 W/cm$^2$ electric power at a 10 nm vacuum gap at a temperature difference of 600 K. Moreover, I also propose a modification to the emitter temperature, which helps us learn about the system's performance in high-temperature situations. By comparing the results above with related works, this work can achieve higher conversion efficiency and electric power after the optimization of the parameters including the thickness and the plasma frequency of the ITO emitter, the chemical potential of the graphene, and the temperature difference. This work promises to be an inspiring guidance for graphene-based theoretical development and further application in NFTPV systems.

**Supplementary Materials:** The following supporting information can be downloaded at: https://www.mdpi.com/article/10.3390/photonics10020137/s1, MATLAB codes.

**Funding:** This research received no external funding.

**Institutional Review Board Statement:** Not applicable.

**Informed Consent Statement:** Not applicable.

**Data Availability Statement:** Not applicable.

**Acknowledgments:** The authors would like to thank the anonymous reviewers for their valuable comments and suggestions.

**Conflicts of Interest:** The authors declare no conflict of interest.

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
