# Peer review of "Enhancement of a Graphene-Based Near-Field Thermophotovoltaic System by Optimization Algorithms and Dynamic Regulations"

_photonics, doi:10.3390/photonics10020137_

Round 1

Author Response

Thanks for the reviewer's efforts to give valuable review comments. The point-by-point response has been uploaded to the attachment.

Reviewer 2 Report

Thermophotovoltaics (TPVs), a heat recovery technique, is faced with low efficiency and power density. It has been proven that graphene helps add new functionalities to optical components and improve their performance for heat transfer. In this work, I study Near-field radiative heat transfer in TPVs based on a composite nanostructure composed of Indium Tin Oxide (ITO) sheet and a narrow bandgap photovoltaic cell Indium Arsenide (InAs). I introduce a new way to calculate nonradiative recombination (NR) and compare the performance of whether the NR is considered. By comparing graphene modulated on the emitter (G-E), on the receiver (G-R), and on both the emitter and the receiver (G-ER), I find the G-ER case can achieve highest current density. However, constrained by the bandgap energy of the cell, this case is far less than the G-E case when 16

it comes to efficiency. By applying Variant Particle Swarm Optimization (VPSO) and dynamic optimization, the model is optimized up to 43.63 % efficiency and 11 W/cm2

electric power at 10 nm vacuum gap with a temperature difference of 600 K.

the manuscript will be accepted if following questions are addressed:

There are good articles in the introduction section, but it is better to mention different fields to make it interesting for the reader. 

My suggestions are?

1.  Khaleel, S.A.; Hamad, E.K.I.; Parchin, N.O.; Saleh, M.B. MTM-Inspired Graphene-Based THz MIMO Antenna Configurations Using Characteristic Mode Analysis for 6G/IoT Applications. Electronics 2022, 11, 2152. https://doi.org/10.3390/electronics11142152

2. Huang, J. ., Wu, Y. ., Su, B. ., & Liu, J. . (2022). Preparation and Electrical Testing of Double Top Gate Graphene Field-Effect Transistor. The Applied Computational Electromagnetics Society Journal (ACES), 37(07), 774–781.https://doi.org/10.13052/2022.ACES.J.370704

3. Beiranvand, B.; Sobolev, A.S. A proposal for a multi-functional tunable dual-band plasmonic absorber consisting of a periodic array of elliptical grooves. J. Opt. 2020, 22, 105005.

4. Beiranvand, B.; Sobolev, A.S.; Sheikhaleh, A. A proposal for a dual-band tunable plasmonic absorber using concentric-rings resonators and mono-layer graphene. Optik 2020, 223, 165587.

5. Xuegang Zhang, Fei Chen, Yijian Jiang, Yinzhou Yan, Lixue Yang, Letian Yang, Xiuhong Wang, Chunlian Yu, Linna Hu, Yuhua Dai, Qiang Wang. Graphene Oxide Modified Microtubular ZnO Antibacterial Agents for a Photocatalytic Filter in a Facial Mask. ACS Applied Nano Materials 2022, 5 (11) , 16332-16343. https://doi.org/10.1021/acsanm.2c03371

In the Schematic diagram of the NFTPV system, explain more about the heat sink and the effect on the mechanism.

Figure 3 and Figure 2 should be merged into one figure and explained more briefly.

Describe Figure 4a separately in the form of a separate figure in the text.

Where is the conclusion of the text?

Compare your work with others

Author Response

(The authors gave the same response as above.)

Round 2

Reviewer 1 Report

The author has addressed all my questions. 

Reviewer 2 Report

all changes are added I accept this manuscript as paper.